# Quorum-sensing synthase mutations re-calibrate autoinducer concentrations in clinical isolates of *Pseudomonas aeruginosa* to enhance pathogenesis

Kayla A. Simanek [1], Megan L. Schumacher[1], Caleb P. Mallery [1], Stella Shen[2], Lingyun Li[3] & Jon E. Paczkowski [1,2] ✉

Quorum sensing is a mechanism of bacterial communication that controls virulence gene expression. *Pseudomonas aeruginosa* regulates virulence via two synthase/transcription factor receptor pairs: LasI/R and RhlI/R. LasR is considered the master transcriptional regulator of quorum sensing, as it upregulates *rhlI/R*. However, clinical isolates often have inactivating mutations in *lasR*, while maintaining Rhl-dependent signaling. We sought to understand how quorum sensing progresses in isolates with *lasR* mutations, specifically via activation of RhlR. We find that clinical isolates with *lasR* inactivating mutations often harbor concurrent mutations in *rhlI*. Using ultra-high-performance liquid chromatography coupled with high-resolution mass spectrometry, we discover that strains lacking *lasR* overproduce the RhlI-synthesized autoinducer and that RhlI variants re-calibrate autoinducer concentrations to wild-type levels, restoring virulent phenotypes. These findings provide a mechanism for the plasticity of quorum sensing progression in an acute infection niche.

*Pseudomonas aeruginosa* is a ubiquitous, opportunistic pathogen that causes tens of thousands of hospital-acquired infections each year[1]. The Centers for Disease Control and Prevention have classified *P. aeruginosa* infections as a serious public health threat due to the prevalence of multidrug-resistant (MDR) strains in healthcare facilities[1]. Moreover, the incidence of hospital-acquired MDR infections increased by 32% between 2019 and 2020 due to the emergence of the COVID-19 pandemic[2]. *P. aeruginosa* persists in healthcare niches by forming biofilms, particularly on medical equipment like catheters and ventilators, which are difficult to eradicate. Consequently, *P. aeruginosa* can establish infections in immunocompromised patients, such as those undergoing cancer treatment or requiring immunosuppressants, as well as patients with underlying pulmonary disorders such as cystic fibrosis (CF)[3–5] and chronic obstructive pulmonary disorders (COPD)[6,7]. These infections can result in chronic pneumonia or life-threatening septic bloodstream infections, resulting in over 2000 deaths per year[1].

*P. aeruginosa* regulates biofilm formation and other virulence traits via quorum sensing (QS). QS is a mechanism of inter-bacterial communication that relies on the production and detection of signaling molecules called autoinducers (AI). In many Gram-negative bacteria, LuxI/R-type synthase/receptor pairs control QS wherein the AI molecule is secreted from the cell and diffuses across the bacterial membrane at high cell density where it binds its cognate cytosolic transcription factor receptor. The receptor-AI complex then alters gene expression to coordinate individualistic and collective behaviors.

[1]Department of Biomedical Sciences, University at Albany, School of Public Health, Albany, New York 12201, USA. [2]Division of Genetics, Wadsworth Center, New York State Department of Health, Albany, New York 12208, USA. [3]Division of Environmental Health Sciences, Wadsworth Center, New York State Department of Health, Albany, New York 12208, USA. ✉e-mail: jon.paczkowski@health.ny.gov

*P. aeruginosa* has three main hierarchical QS circuits: (1) the Las system, (2) the Rhl system, and (3) the Pqs system[8,9]. The Las system consists of the LuxI family AI synthase LasI, which synthesizes the acyl homoserine lactone (AHL) N-(3-oxododecanoyl)-L-homoserine lactone (3OC$_{12}$HSL), and the LuxR-type receptor LasR. When bound by 3OC$_{12}$HSL, LasR dimerizes and is an active transcription factor that upregulates the *rhl* system[10-12], among other virulence genes. Homologous to the Las circuit, the Rhl system consists of the AI synthase/receptor pair RhlI/R. RhlI synthesizes N-butanoyl-L-homoserine lactone (C$_4$HSL)[13], which binds to and activates the transcription factor receptor RhlR. RhlR upregulates the expression of numerous virulence factors such as the genes involved in phenazine biosynthesis (with the final product being 5-methyl-1(5H)-phenazinone−pyocyanin) and rhamnolipid production. The third system, the *Pseudomonas* quinolone system, is upregulated by LasR but repressed by RhlR[14]. The *pqsABCDE* operon encodes enzymes responsible for the synthesis of the AI molecule 2-heptyl-3-hydroxy-4(1H)-quinolone (PQS), which binds the receptor PqsR resulting in a positive feedback loop[15,16]. We have previously shown that PqsE, a metallo-β-hydrolase enzyme, physically binds to RhlR[17] and enhances the affinity of RhlR for its target promoters independent of its catalytic activity[18]. Moreover, both C$_4$HSL and PqsE binding are required for maximal RhlR activity[19]. However, recent research suggests that QS in clinical strains of *P. aeruginosa* does not follow the canonical model described above but signaling via RhlI-RhlR-PqsE remains intact.

The high-traffic environment of a hospital and the selective pressure of antibiotic usage drive pathogen evolution in healthcare settings. Clinical strains of *P. aeruginosa* often evolve *lasR* inactivating mutations during chronic infection[20,21], leading to a re-wiring of the QS signaling network. These strains have been termed social cheaters as it is hypothesized that *lasR* mutations repress QS to defray the cost of synthesizing energetically expensive metabolites[22]. Instead, these strains achieve a fitness advantage by leeching public goods from wild-type (WT) bacteria[23]. Another hypothesis is that LasR- strains are better adapted to the lung environment because these mutants grow better in a chronic infection niche[24]. These strains maintain QS through the Rhl system despite harboring nonfunctional LasR variants (LasR-)[23]. Previous studies demonstrated that the passage of LasR- strains results in mutations in *pqsR*[22,25,26], the receptor/transcription factor for the Pqs system. Another study showed that LasR- strains evolve mutations in *mexT*[27,28], a regulator of the MexEF-OprN efflux system, which secretes C$_4$HSL[29]. Together, these data suggest that RhlR activation converges on C$_4$HSL in LasR- clinical strains given that (1) *pqsR* inactivating mutations would preclude RhlR activation by *pqsE*[18,19] (2) *mexT* mutations would increase C$_4$HSL extrusion and cell-cell signaling[29] and (3) LasR- strains do not respond to 3OC$_{12}$HSL[24]. Indeed, the levels of C$_4$HSL are important for *P. aeruginosa* to establish acute infections[30]. In the cohort of *P. aeruginosa* clinical strains analyzed in this study, *pqsE, rhlR*, and *lasI* single nucleotide polymorphisms (SNPs) are infrequent and often silent. Conversely, *rhlI* SNPs are frequent and consistent (Supplementary Table S1), suggesting that these acute infection strains have evolved *rhlI* mutations in response to *lasR* inactivating mutations.

Here, we investigate the function of three clinically evolved, nonsynonymous amino acid changes in RhlI: G62S, D83E, and P159S. We show that these clinically evolved RhlI variants restored pyocyanin production in a Δ*lasR* strain. We use ultra-high-performance liquid chromatography (UHPLC) coupled with high-resolution mass spectrometry (HRMS) to directly detect and analyze the levels of AHL AI synthesized by RhlI variants. We show that RhlI variants re-calibrate the concentrations of extracellular C$_4$HSL, resulting in increased virulence factor production in a Δ*lasR* background. We hypothesize that LasR- clinical strains evolve compensatory mutations in *rhlI* to optimize the levels of AI for RhlR activation to drive the expression of genes involved in pathogenesis.

## Results

### Identification of *rhlI* SNPs in clinical strains of *P. aeruginosa*

We analyzed whole genome sequencing data of 56 *P. aeruginosa* clinical isolates (Supplementary Table S1) from the Wadsworth Center, New York State Department of Health to identify SNPs in QS genes (NCBI BioProject PRJNA288601). We found that *rhlR, pqsE* and *lasI* were WT in their amino acid sequences compared to PA14, while *lasR* (Supplementary Table S1) and *rhlI* had numerous SNPs resulting in amino acid substitutions. *lasR* mutations in clinical strains of *P. aeruginosa* have been well documented. The mutations are stochastic and often result in non-functional proteins[20]. The *rhlI* SNPs we discovered were frequent and consistent, suggesting a functional role in the maintenance of these specific mutations. We chose three SNPs of interest that resulted in the following amino acid changes in RhlI: G62S, D83E, and P159S (Fig. 1a). We used AlphaFold[31] to map the location of the variants on the structure of RhlI relative to the substrate binding pockets using homology modeling to the two experimentally determined LuxI synthase structures: LasI[32] from *P. aeruginosa* and EsaI[33] from *Pantoea stewartii* (Fig. 1b). The predicted structure aligns well with the experimentally determined structures from other systems as RhlI has a typical α-β-α fold that results in a deep cleft to allow for binding to the *S*-adenosylmethionine and acyl-acyl carrier protein substrates. While the structures of LasI and EsaI were not determined bound to substrate, the structure of the two binding pockets are similar to that of the GNAT family of acetyltransferases, which carry out a similar acylation reaction[34]. Thus, we are confident in the fold of the computationally derived structure of RhlI and the location of the variant residues therein, as it superimposes with LasI with a root mean squared deviation of approximately 2.4 Å (Supplementary Fig. S1a). The residues G62, D83, and P159 are peripheral to the core anti-parallel β-sheet fold in RhlI, indicating that they are not directly involved in binding substrate or catalysis; residues G62S and P159 flank a β-sheet predicted to be the acyl-binding site of Rhl and residue D83 is located in a loop adjacent to the active site[35] (Fig. 1b). Conservation scores mapped on the PA14 RhlI structure using sequences from our cohort of 56 clinical strains revealed that these three sites were the most variable. Furthermore, the mutations of interest characterized here have been identified by other studies[23], and are consistent within the cohort of isolates we analyzed, suggesting that there is a functional advantage to maintaining these particular mutations (Fig. 1c). Additionally, these sites were poorly conserved among the 150 LuxI synthase homologs sequences we examined by conservation analyses (Fig. 1d, Supplementary Fig. S1b), indicating that these residues can be variable and still maintain functionality across a diverse range of homologs.

To test whether the three RhlI variants were functional AI synthases, we recombined *rhlI* mutants of interest into our PA14 laboratory strain. We then performed a supernatant assay in which the cell-free supernatants of strains harboring RhlI variants were supplemented to a Δ*rhlI* recipient strain. We used pyocyanin production as a proxy for RhlI functionality because pyocyanin is dependent on the C$_4$HSL activation of RhlR. Thus, a Δ*rhlI* recipient would not produce pyocyanin unless supplemented with sufficient concentrations of C$_4$HSL from the cell-free supernatant. Indeed, supplementation of a Δ*rhlI* strain with Δ*rhlI* strain cell-free supernatant did not result in pyocyanin production (Fig. 1e). Conversely, all three of the cell-free supernatants from strains expressing RhlI variants induced pyocyanin production in the Δ*rhlI* recipient strain comparable to WT levels, confirming that all three RhlI variants are functional synthases (Fig. 1e).

### Strains carrying RhlI variants compensate for *lasR* signaling deficiency to produce pyocyanin

The frequency and consistency of *rhlI* SNPs in clinical *P. aeruginosa* isolates (Fig. 1a and Supplementary Fig. S2) led us to hypothesize that they could be compensatory mutations for non-functional *lasR* mutations. Indeed, these SNPs were observed in clinical isolates from

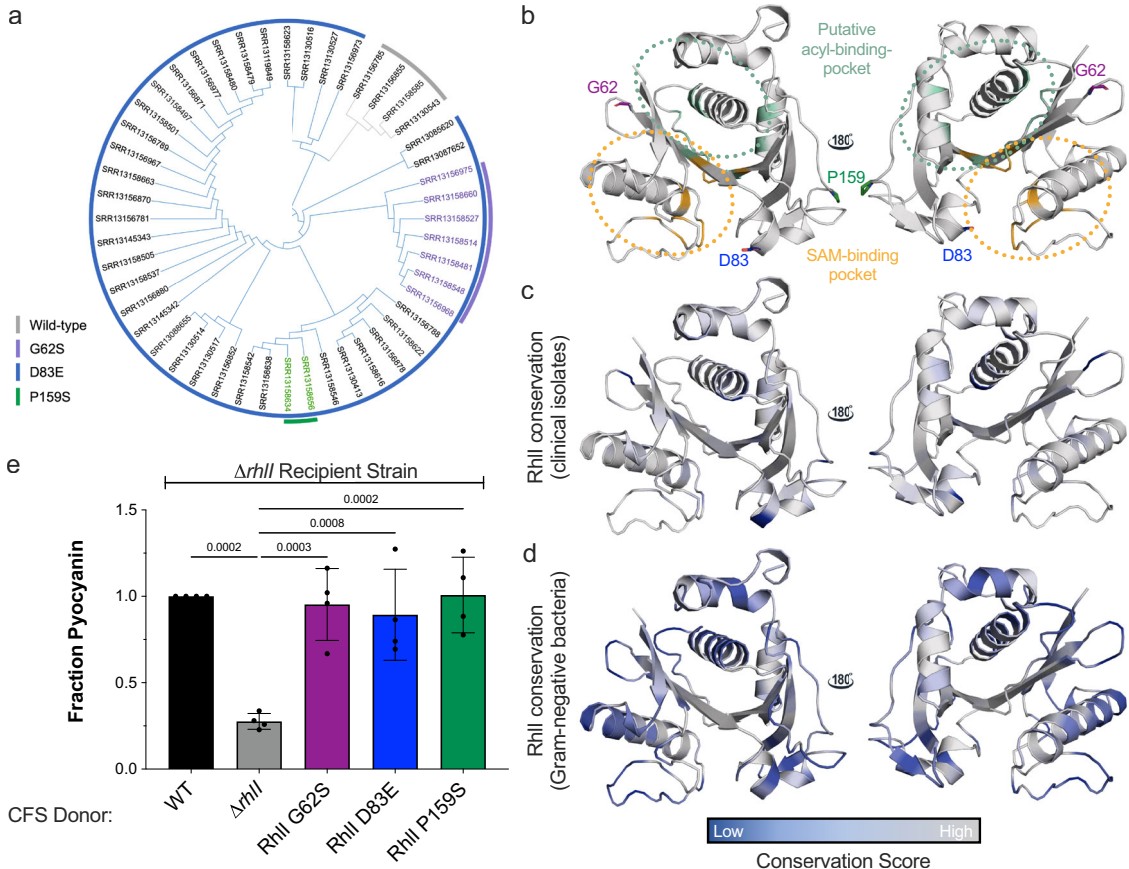

**Fig. 1 | Identification of RhlI variants in clinical *P. aeruginosa* strains.**
**a** Dendrogram of clinical isolates based on *rhlI* sequences. Strains containing SNPs are colorized to highlight RhlI variants of interest: G62S is purple; D83E is blue; P159S is green. **b** The predicted structure of RhlI was determined using AlphaFold[31]. The putative acyl-chain binding site is highlighted in light green. Residues predicted to be involved in catalysis are shown in orange. The putative acyl-binding pocket and catalytic residues were assigned based on the existing experimentally solved structures of RhlI homologs LasI (PDB: 1RO5)[32] from *P. aeruginosa* and EsaI (PDB: 1KZF)[33] from *P. stewartii*. The location of residues altered by the *rhlI* SNPs G62 (purple) D83 (blue), and P159 (green) are indicated and depicted in stick form. ConSurf maps depicting residue conservation in **c** clinical strains and **d** 150 LuxI synthase homologs. For both ConSurf maps, conservation was mapped on a scale of 1–9 with low conservation scores (variable residues) shown in blue and high conservation scores (conserved residues) shown in gray. **e** Cell-free supernatant (CFS) supplementation assay to monitor pyocyanin production in a Δ*rhlI* recipient strain after 8 h of growth with supernatants from WT (black), Δ*rhlI* (gray), RhlI G62S (purple), RhlI D83E (blue), and RhlI P159S (green) variant strains. Supplementation of WT cell-free supernatants to the Δ*rhlI* recipient was set to 1 in every biological replicate and mutant strains were normalized to that value. Bars represent the mean of four biological replicates. Error bars represent standard deviations of the means of biological replicates. Statistical analyses were performed using a one-way ANOVA with Dunnett's multiple comparisons test with the Δ*rhlI* donor strain set as the control.

other studies, suggesting that their prevalence in our cohort of isolates was not simply due to epidemiological constraints and supporting the hypothesis that *rhlI* SNPs are maintained in clinical populations because they confer a fitness advantage. To determine if these RhlI variants compensated for a loss-of-function *lasR*, we introduced each SNP at the *rhlI* locus in a PA14 Δ*lasR* background. We then performed a time course assay in phosphate-limiting media[36] and monitored the production of pyocyanin. Both the growth ($OD_{600\,nm}$) and pyocyanin ($OD_{695\,nm}$) were measured at four different time points over the course of a 24-h period (Fig. 2a–d). We found that a Δ*lasR* strain had a pyocyanin production lag, but that the introduction of RhlI variants restored pyocyanin production in a Δ*lasR* background to WT levels. Furthermore, the RhlI variants did not dramatically alter the timing of pyocyanin production in a WT background, suggesting that these variants specifically aid in maintaining QS in strains lacking a functional Las system (Fig. 2c, d). All Δ*lasI* background (Supplementary Fig. S3A–d) strains produced pyocyanin at a similar rate as WT, and RhlI variants did not confer increased pyocyanin production in this background, indicating that there is a decoupling of the *las* system as it relates to its regulation of the *rhl* system.

To determine if this phenomenon was specific to pyocyanin production or general to other RhlR-dependent traits, we tested the swarming phenotypes of WT, Δ*lasR*, Δ*rhlI*, and Δ*lasR* strains harboring RhlI variants on semi-soft agar media (Fig. 2e). Swarming motility is controlled by RhlR via positive regulation of the *rhlA* promoter in a $C_4HSL$-dependent manner[37,38]. Thus, we expected that RhlI variants would restore swarming in a Δ*lasR* strain[39]. Indeed, all Δ*lasR* strains harboring RhlI variants restored swarming motility to WT levels (Fig. 2e). These data support the hypothesis that RhlI variants are compensatory mutations and restore virulence factor production in strains lacking functional LasR.

**Strains expressing RhlI variants have altered AI production**
To investigate the mechanism of RhlI variant restoration of pyocyanin production in a Δ*lasR* background, we determined the concentrations of metabolites in the cell-free supernatant from the same time course cultures in the pyocyanin production assay described in Fig. 2 using ultra-high-performance liquid chromatography (UHPLC) coupled with high-resolution mass spectrometry (HRMS)[40,41]. Absolute concentrations of $C_4HSL$ were normalized to $OD_{600}$ at each time point. We found

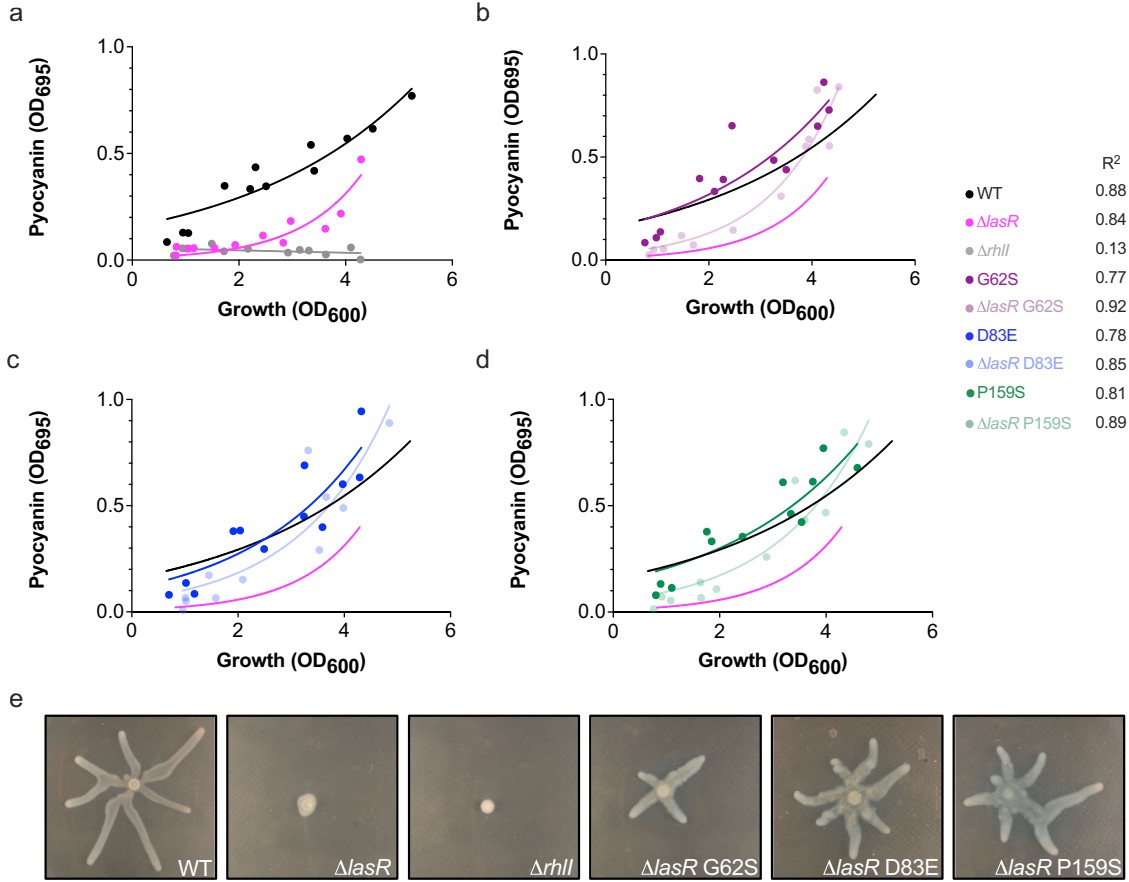

**Fig. 2 | RhlI variants restore pyocyanin production in a ΔlasR background.**
**a** Summary graph of the pyocyanin time course experiment for WT (black), ΔlasR (magenta) and ΔrhlI (gray) strains. Measurements were taken at four time points over a 24-h period (approximately 5, 10, 18 and 24 h). Time course pyocyanin production assay for: **b** RhlI G62S variant data in WT background (purple) and ΔlasR backgrounds (light purple); **c** RhlI D83E variant data in WT (blue) and ΔlasR backgrounds (light blue); **d** RhlI P159S variant data in WT (green) and ΔlasR backgrounds (light green). One measurement was made per strain per time point. The experiment was performed in triplicate, resulting in the 12 total data points depicted in the graph. Non-linear regression analyses were performed for every genotype and the best fits shown; best fit lines for WT (black) and ΔlasR (magenta) control strains were plotted and copied for reference in subsequent panels for the RhlI variant strains. **e** Representative images of WT, ΔlasR, ΔrhlI, and ΔlasR RhlI variant swarming phenotypes.

that a ΔlasR strain produced 2-fold more C$_4$HSL than WT at later time points (Fig. 3a–d). Surprisingly, the RhlI variants attenuated C$_4$HSL production in a ΔlasR background after the second time point, reducing the C$_4$HSL concentrations to levels observed in the WT strain. We note that a similar profile for C$_6$HSL was observed, albeit at lower concentrations (Supplementary Fig. S4a, b). Conversely, mass spectrometry data of a ΔrhlI strain showed increased levels of 3OC$_{12}$HSL (Supplementary Fig. S4c, d), suggesting that the pool of shared precursor substrates, namely the acylated acyl carrier protein and S-adenosylmethionine (Fig. 1b) required for the AHL condensation reaction, is funneled into the opposing biosynthetic pathway when one branch of the signaling cascade is disrupted. While we observed an initial burst in C$_4$HSL production in a ΔlasI strain at the second time point, production decreased and leveled off in subsequent time points (Supplementary Fig. S4e), unlike in a ΔlasR strain, which maintained elevated levels of C$_4$HSL throughout the time course (Fig. 3).

Overall, the levels of C$_4$HSL and C$_6$HSL inversely correlated with the time points at which robust pyocyanin production was observed in these strains (Fig. 2a–d, Supplementary Fig. S4A,B). These results seemed counterintuitive, as higher C$_4$HSL concentrations are often associated with the upregulation of virulence phenotypes. To confirm that additional factors in the cell-free supernatants were not obscuring our pyocyanin readings, we performed mass spectrometry analysis to detect pyocyanin, using a 5-methyl-1(5H)-phenazinone (pyocyanin) as a standard. We found that pyocyanin concentrations (Supplementary

Fig. S5A–d) correlated with our absorbance readings in Fig. 2 and Supplementary Fig. S3. In total, these results showed that high concentrations of C$_4$HSL suppressed RhlR-dependent phenotypes. This was most readily observable in the transition between the second and third time point; strains containing RhlI variants in a ΔlasR background had a decline in C$_4$HSL concentrations and a concomitant increase in pyocyanin production. We tested if high concentrations of C$_4$HSL could repress pyocyanin production in vivo using a dose-response assay (Fig. 3e). Pyocyanin production in a ΔrhlI and WT strain was repressed at 10 µM C$_4$HSL, while levels remained constant in a ΔlasR strain. Therefore, we hypothesized that an optimal concentration of C$_4$HSL is required for RhlR activation in vivo and that clinically evolved RhlI G62S, D83E, and P159S variants reduce the efficiency of the RhlI synthase to re-calibrate C$_4$HSL levels in a ΔlasR background to appropriately activate RhlR-dependent signaling.

**RhlI variants synthesize the optimal concentration of C$_4$HSL for RhlR activation**
The expression of RhlR target genes has different C$_4$HSL dependencies[42]. We hypothesized that the reduced levels of C$_4$HSL in RhlI variant strains would impact the expression of C$_4$HSL-sensitive genes. We analyzed the expression of two genes with different C$_4$HSL dependencies: hydrogen cyanide (hcnA) is C$_4$HSL independent and rhamnolipid synthase (rhlA) is C$_4$HSL dependent. We performed qRT-PCR on cell pellets from the time course experiment described in

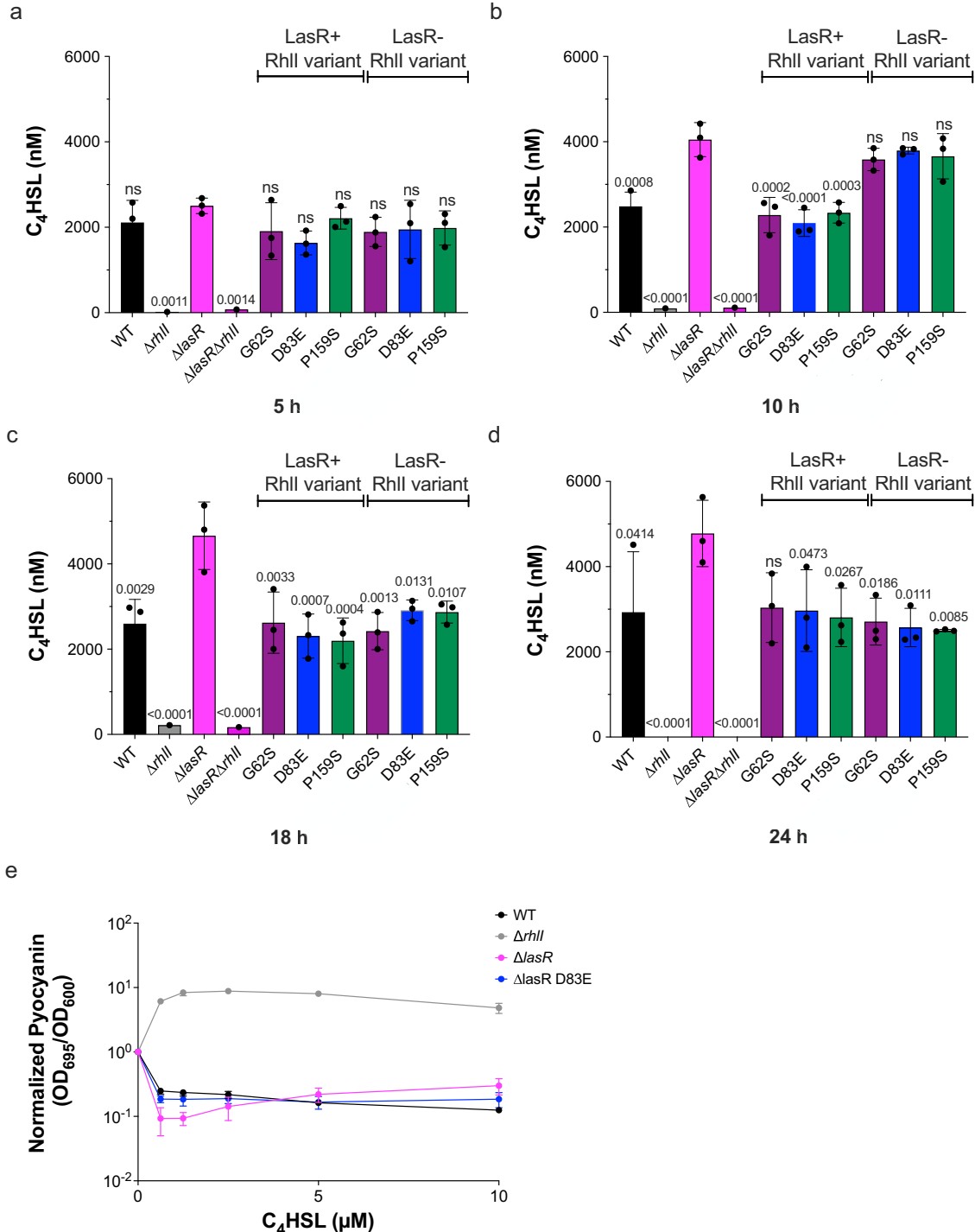

**Fig. 3 | RhlI variants reduce the levels of C$_4$HSL in a Δ*lasR* background.** Absolute concentrations of C$_4$HSL from cell-free supernatants of strains from the time course cultures used in Fig. 2 were measured using ultra-high-performance liquid chromatography (UHPLC) coupled with a high-resolution mass spectrometer (HRMS) after **a** 5, **b** 10, **c** 18, and **d** 24 h of growth. Bars represent the mean of three biological replicates. Error bars represent standard deviations of the means of biological replicates. Statistical analyses were performed using a one-way ANOVA with Dunnett's multiple comparisons test with the Δ*lasR* strain set as the control. **e** Pyocyanin production in a C$_4$HSL dose-response assay of WT (black), Δ*rhlI* (gray), and Δ*lasR* (magenta) strains. Bars represent the mean of three biological replicates. Error bars represent standard deviations of the means of biological replicates.

Fig. 2. For brevity, we performed these experiments with the RhlI D83E variant because all three variants altered C$_4$HSL levels similarly in our mass spectrometry data (Fig. 3) and we expected changes in gene expression to be comparable across the variants. We found that the expression of *rhlA* was dynamic and heavily influenced by changes in the concentrations of C$_4$HSL. The expression of *rhlA* plateaued in a

Δ*lasR* strain and declined in a WT strain at the final time point, indicating a cessation of virulence factor production at high cell density (Fig. 4a). We note that a similar effect was observed for pyocyanin production in end-point cultures (Fig. 2). The expression of *rhlA* in a Δ*lasR* D83E strain was delayed but increased 4-fold by the final time point (Fig. 4a) and this was not due to the upregulation of either *rhlI* or

*rhlR* (Fig. 4b, c). As expected, we found that *hcnA* expression was generally unchanged across strains (Fig. 4d). The expression of *hcnA* by RhlR is known to be heavily dependent on its interaction with PqsE and the modest decrease in *hcnA* expression is likely due to the RhlR:C$_4$HSL-dependent repression of the *pqsABCDE* operon at high cell density. Taken together, these data suggested that RhlI variants synthesized optimal concentrations of AHL required for RhlR activation and subsequent virulence gene expression in vivo.

We then performed qRT-PCR to determine if deletion of *lasI* resulted in transcription trends similar to a Δ*lasR* strain. Interestingly, we found that both *rhlI* and *rhlR* transcript levels increased over time in the Δ*lasI* strain (Supplementary Fig. S6a, b), unlike the Δ*lasR* strain (Fig. 4b, c). As expected, expression of *hcnA* was static in both the Δ*lasI* and Δ*lasI* D83E strains (Supplementary Fig. S6c), and *rhlA* expression increased steadily over time (Supplementary Fig. S6d), consistent with the increased levels of *rhlI* and *rhlR*. These results suggested that LasR upregulates *rhlR* independently of its activating ligand 3OC$_{12}$HSL and supports our findings of the decoupling of the *las* system as it relates to regulating the *rhl* locus (Supplementary Fig. S3).

## Phenotypic and transcriptional profiling of clinical isolates reveals a dependency on optimal C$_4$HSL levels
To determine if the RhlI variants that restored virulence phenotypes in recombinant Δ*lasR* laboratory strains had the same effect on virulence factor production in clinical strains, we selected six clinical isolates from our cohort of strains (Supplementary Table S1) based on the representation of genotypic clustering (Fig. 1a). For our cohort of

isolates, the genotype frequencies for *lasR* and *rhlI* are as follows: 8.5% WT *lasR* and WT *rhlI*, 30% WT *lasR* and mutant *rhlI*, 62% mutant *lasR* and *rhlI*. It should be noted that no isolate in the cohort has a mutant *rhlI* and WT *lasR*. We determined their C$_4$HSL and pyocyanin production profiles via HRMS (Supplementary Fig. S7a). These data were largely consistent with our model, showing optimal C$_4$HSL concentrations leading to robust pyocyanin production. Strains 680, 685, and 687 overproduced C$_4$HSL, which led to little or no pyocyanin production in strains 680 and 687 (Supplementary Fig. S7a, b). We note that strain 685 was an outlier in our data, as it overproduced C$_4$HSL but still synthesized high levels of pyocyanin. We address this observation below. Conversely, strain 682 produced ~5 μM C$_4$HSL, which, in our estimation, is the optimal concentration of C$_4$HSL production (Supplementary Fig. S7a). Indeed, strain 682 produced more pyocyanin than any other strain (Supplementary Fig. S7a, b). Strains 683 and 686 both produced between 2 and 3 μM C$_4$HSL, which resulted in lower and more variable levels of pyocyanin compared to strain 682 (Supplementary Fig. S7a). This is likely because 683 and 686 have WT *lasR* with RhlI variants (Supplementary Table S1), and the levels of C$_4$HSL are suboptimal for RhlR activation.

To follow up on these trends with additional phenotypic and transcriptional analyses, we selected specific clinical strains based on their genotype: 680 has WT *lasR* and *rhlI* and 682 and 685 have mutant *lasR* and *rhlI* (Supplementary Table S1). To determine if higher concentrations of C$_4$HSL repressed pyocyanin production in clinical strains similar to what we observed in Fig. 3e, we performed a dose-response assay (Supplementary Fig. S7c). These experiments

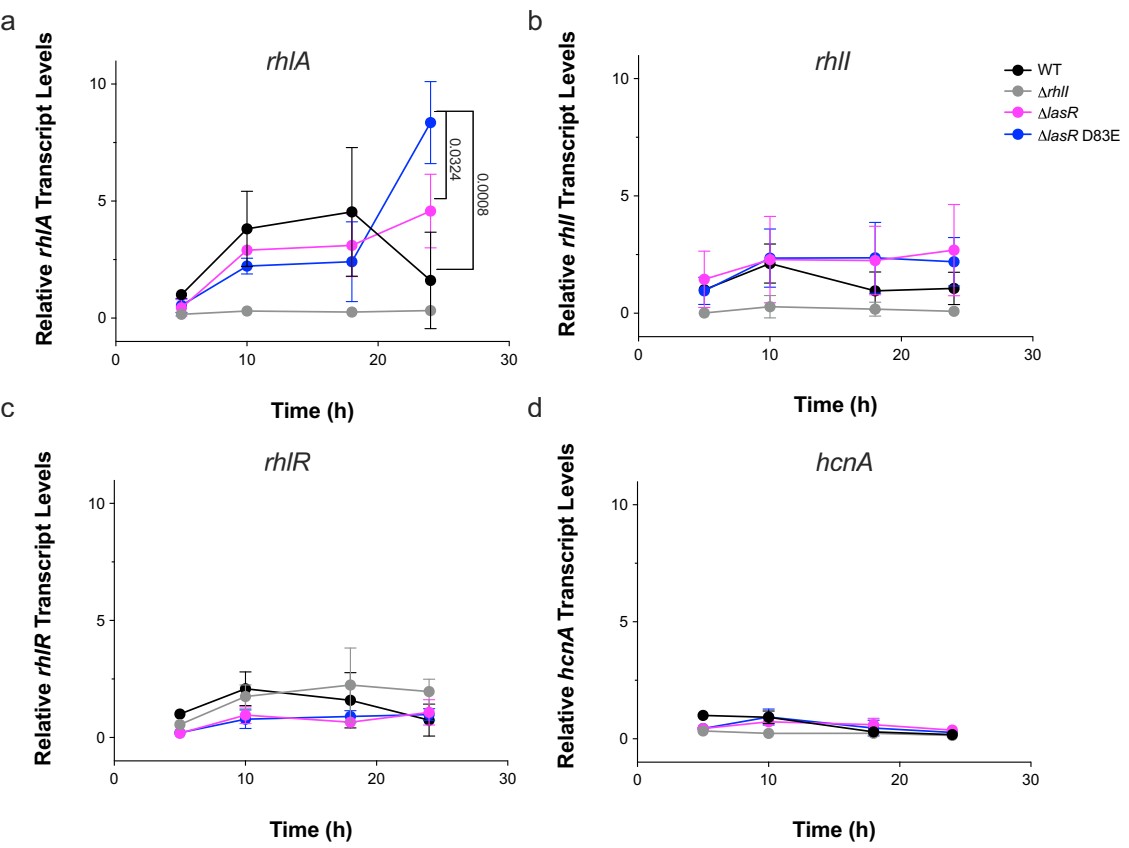

**Fig. 4 | RhlI variants optimize C$_4$HSL concentrations to increase virulence gene transcription.** The relative transcript levels of **a** *rhlA*, **b** *rhlI*, **c** *rhlR*, and **d** *hcnA* in WT (black), Δ*rhlI* (gray), Δ*lasR* (magenta), and Δ*lasR* RhlI D83E (blue) strains. *gyrA* was used as an endogenous control for qRT-PCR experiments and data was normalized to WT PA14 expression levels at the 5-h time point. Bars represent the mean of three biological replicates performed in technical duplicate. Error bars represent standard deviations of the means of biological replicates. Statistical analyses were performed using multiple paired *t*-tests using the two-stage step-up method between Δ*lasR* and Δ*lasR* D83E or Δ*lasR* D83E and WT with a *p*-value discovery cutoff of 0.05. Only *p*-values < 0.05 are shown.

revealed that pyocyanin production in strains 680 and 682 can be repressed when exogenous $C_4HSL$ is added during growth. However, strain 685 produced pyocyanin in a dose-dependent manner and, at the concentrations we tested, we did not observe an inhibitory effect of $C_4HSL$ on pyocyanin production, indicating that this isolate is insensitive to the repressive effects of $C_4HSL$. The most glaring genotypic difference between strain 685 and all other strains in the cohort is the mutation in *lasR*. The *lasR* gene in 685 has an internal, in-frame deletion that results in a 34 amino acid truncation but is otherwise intact (Δ81-115) (Supplementary Table S1). The *lasR* gene in strain 682 encodes a LasR variant only 80 amino acids long (Supplementary Table S1). Therefore, 685 might still be able to respond to $3OC_{12}HSL$ or its LasR variant possesses a non-canonical function that we cannot account for, making its QS network responsive to higher levels of AI. This is particularly relevant within the context of the decoupling of the *las* system that we previously observed (Supplementary Fig. S3). We next performed qRT-PCR on clinical strains 680, 682, and 685 and found that *rhlR* transcript levels were relatively consistent between strains, whereas *rhlA* expression levels were similar to or greater than those of a Δ*lasR* strain (Supplementary Fig. S7d), which mirrors our results in Fig. 4a. We note that the Δ*lasR* parent strain is PA14, which is genotypically very different than the clinical strains. Taken together, these data show that the expression of virulence phenotypes in acute infection isolates is largely controlled by the levels of $C_4HSL$.

### LasR- strains expressing RhlI variants are more pathogenic than a LasR- in an infection model

To determine if RhlI variants were more pathogenic in an in vivo model, we performed a fast kill assay with a *Galleria mellonella* moth larvae infection model[43]. The moth larvae were infected with $10^6$ cells of *P. aeruginosa* and monitored for 10 h. The death of the larvae was determined by melanization and failure to respond to physical probing. The Δ*lasR* strain was the least virulent and failed to kill all the larvae by the 10-h time point, while a Δ*rhlI* strain was the most virulent, consistent with other animal model experiments[42]. This suggests that decreased $C_4HSL$ signaling increases the virulence of *P. aeruginosa*, but overproduction of $C_4HSL$ is detrimental to the progression of pathogenesis, consistent with all other traits we monitored in this study. Indeed, Δ*lasR* strains harboring RhlI variants were more virulent than the Δ*lasR* strain and killed larvae at a rate comparable to the Δ*rhlI* (Fig. 5a). These results suggest that RhlI variants evolve in clinical strains of *P. aeruginosa* to enhance or re-establish virulence of LasR- strains during infection.

## Discussion

QS in clinical strains of *P. aeruginosa* does not occur as it has been canonically described in laboratory PA14 or PAO1 strains. Frequent inactivating mutations in the QS master transcriptional regulator *lasR* have been reported, and these strains tend to evolve compensatory mutations in other QS-related genes to restore group behavior signaling. Here, we identified SNPs in the *rhlI* gene of clinical isolates that altered the function of RhlI to compensate for the loss of LasR function. Specifically, we showed that RhlI variants synthesize lower levels of $C_4HSL$ and that the reduction of $C_4HSL$ concentrations in a Δ*lasR* background results in the restoration of virulence expression both in vitro and in vivo. Meta-bromo-thiolactone (mBTL) is a small molecule agonist of RhlR that partially inhibits its regulatory function at high concentrations[44]. Similarly, we anticipate that the loss of *lasR* funnels the pool of acyl homoserine lactone precursor molecules into the $C_4HSL$ biosynthetic pathway, resulting in $C_4HSL$ levels that repress RhlR instead of activating it. While LasR- clinical isolates of *P. aeruginosa* possess a fitness advantage over LasR+ strains, either by social cheating[22] or enhanced fitness in a lung niche[22], these strains are rendered less virulent. We purport that LasR- clinical strains evolve

secondary mutations in *rhlI* to enhance virulence (Fig. 5b). The *rhlI* mutations characterized in this study (D83E, G62S, and P159S) have been identified by other studies, suggesting that the evolution of these mutations is not an epidemiological constraint of our cohort of isolates. Therefore, the results of this study are applicable to widely circulating modern clinical strains of *P. aeruginosa*.

While it was unexpected that the *lasI* and *lasR* deletions had different effects on *rhlR* expression, it is not a novel observation, as other studies have shown that Δ*lasR* and Δ*lasI* strains have different phenotypes and pathogenicity[45,46]. However, the discrepancy in the progression of $C_4HSL$ production over time in Δ*lasR* and Δ*lasI* backgrounds might explain why loss-of-function mutations in *lasI* are not readily observed in clinical strains. While we are unsure of the mechanism of the decoupling of *lasI* and *lasR* behaviors, it does support our findings from Fig. 2 and Supplementary Fig. S3a, which showed that the time-course dependency of pyocyanin production in Δ*lasR* and Δ*lasI* strains were different. Furthermore, it supports our conclusions that the concentration of $C_4HSL$ is the main driver of signaling in these backgrounds. This is also evidenced by increased *rhlR* transcription in a Δ*lasI* strain but not in a Δ*lasR* background (Supplementary Fig. S6b), which suggests that unliganded LasR can act as a transcription factor. It was recently reported that $3OC_{12}HSL$ accumulates in LasR- strains grown on a solid surface[47], suggesting that $3OC_{12}HSL$ has an alternative role in QS other than activating LasR.

We note that RhlI variant strains produce similar levels of $C_4HSL$ in both LasR+ and LasR- backgrounds and that those levels are comparable to a WT strain. If RhlI variants were catalytically deficient, it would be expected that a $C_4HSL$ deficiency would also be observed in a LasR+ strain. The RhlI variants studied here are not located in the predicted catalytic site of RhlI[48]. It is possible that the *rhlI* mutations identified in this study simply reduce the affinity of RhlI for acyl-chain-carrying substrates, thereby reducing enzyme efficiency. In a WT background, LasI competes with RhlI for shared precursors. Mass spectrometry data of $3OC_{12}HSL$ levels showed that a Δ*rhlI* strain produced high levels of $3OC_{12}HSL$, and vice versa (Supplementary Fig. S4c), suggesting that the ratio of $3OC_{12}HSL:C_4HSL$ is important for the activation of RhlR. We expect that in a LasR- strain, where *lasI* is not upregulated and no $3OC_{12}HSL$ is produced, *rhlI* mutants evolve to reduce the binding affinity of RhlI to re-calibrate the levels of $C_4HSL$ synthesized in a LasR-background. This would be beneficial in a LasR- background in which the pool of shared acyl homoserine lactone precursors overwhelm the $C_4HSL$ biosynthetic pathway.

Our data revealed that the ratio of $C_4HSL$ concentration to cell density, rather than $C_4HSL$ levels alone, dictates the timing of virulence factor production in clinical strains. We hypothesize that the loss of LasR inadvertently prevents the activation of RhlR by increasing $C_4HSL$ concentrations to an inhibitory level reducing virulence. High concentrations of $C_4HSL$ at low cell density inhibit RhlR, whereas attenuated $C_4HSL$ levels are appropriate for the activation of cytosolically available RhlR. It should also be noted that it is still unclear how the *rhl* system is upregulated in clinical LasR- strains, and this is an avenue of research that we are pursuing. Furthermore, we hypothesize that SNPs in *rhlI* evolved in clinical strains of *P. aeruginosa* to compensate for the loss of *lasR* by restoring the balance of $C_4HSL$ synthesis, resulting in optimal concentrations for RhlR activation and delayed but robust virulence factor expression (Fig. 5b). This could be particularly important for the adaptability of LasR-strains[21], and could mediate reversion from a chronic phenotype to cause new infections, as LasR- strains are deficient in establishing acute infections[30,45,46].

## Methods

The research conducted in this study complies with all regulations set forth by the Wadsworth Center Institutional Review Board.

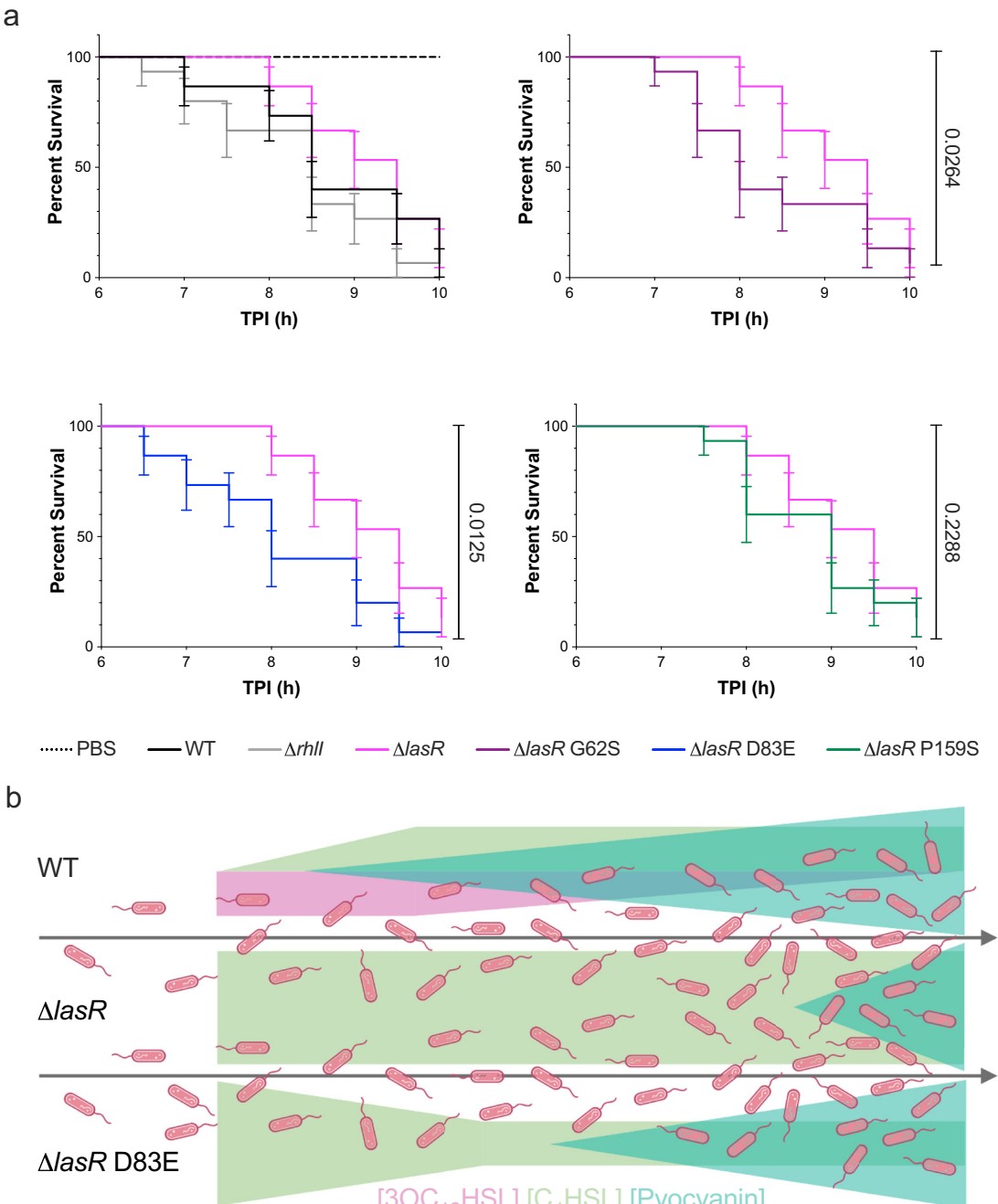

**Fig. 5 | Strains expressing RhlI variants enhance pathogenesis in a Δ*lasR* background. a** Survival assay of *G. mellonella* larvae challenged with WT (black), Δ*rhlI* (gray), Δ*lasR* (magenta), Δ*lasR* G62S (purple), Δ*lasR* RhlI D83E (blue), Δ*lasR* P159S (green) over the course of a 10-h infection. The dotted black line represents a PBS blank injection. TPI time post-infection in hours. Experiments were performed in biological triplicate with 5 larvae per strain. All strains for each survival assay replicates were performed at the same time and plotted on separate graphs for clarity. Error bars represent the standard error of the mean. Statistical analyses of the larval survival curves for Δ*lasR* compared to Δ*lasR* RhlI variant strains were performed using the Gehan-Breslow-Wilcoxon test. **b** Model for RhlI variant restoration of virulence factor expression by modulation of C4HSL (green) and 3OC12HSL (pink) levels in clinical strains of *P. aeruginosa*. Δ*lasR* strains synthesize inhibitory levels of C4HSL early, resulting in the delayed production of virulence factors like pyocyanin (blue triangles). RhlI variants (i.e., D83E) produce less C4HSL, restoring pyocyanin production in a Δ*lasR* background. The figure was made with BioRender.

## Cell-free supernatant supplementation assay

Donor and recipient strains were cultured from freezer stocks in 3 mL LB overnight at 37 °C with shaking. High-cell density recipient cultures were back-diluted 1:10 (500 μL) in 4 mL FDS- media (2% glycerol, 10 g/L DL-alanine, 50 μM iron (III) citrate, 0.1 M Na₂SO₄, 20 mM MgCl₂, 500 μM K₂HPO₄)[49]. High-density donor cultures were pelleted, and 500 μL cell-free supernatant was supplemented to the back-diluted recipient culture. Cultures were incubated for 5 h at 37 °C with shaking. Growth was measured at OD$_{600}$ (1:10 dilution) and pyocyanin at OD$_{695}$. Pyocyanin production was calculated using the equation below.

$$\text{Fraction pyocyanin} = \frac{\left(\frac{\text{OD}_{695}}{\text{OD}_{600}\times 10}\right)\text{mutant}}{\left(\frac{\text{OD}_{695}}{\text{OD}_{600}\times 10}\right)\text{wild type}} \quad (1)$$

## Pyocyanin time course assay

Strains were cultured from freezer stocks in 3 mL LB overnight at 37 °C with shaking. High-cell density cultures ($OD_{600}$ ~ 2.0) were then back-diluted 1:10 in 5 mL FDS- media and incubated for 24 h. Time points were collected at 5, 10, 18, and 24 h. 1 mL of culture was collected at each time point and growth was measured by diluting the culture 1:10 in 1 mL FDS- media and reading the absorbance at $OD_{600}$. The remainder of the 1 mL sample was pelleted, and the absorbance of cell-free supernatant was measured at $OD_{695}$ to quantify pyocyanin. Cell pellets and 100 μL of cell-free supernatant were frozen for RNA extraction and mass spectrometry, respectively.

## Swarming motility assay

Swarming assays were performed as described previously[39]. Briefly, cultures of *P. aeruginosa* PA14 and the mutants were grown overnight in LB. 2 μL of the stationary phase cultures were spotted onto swarming agar medium (1 mM $MgSO_4$, 8.6 mM NaCl, 1 mM $CaCl_2$, 20 mM $NH_4Cl$, 22 mM $KH_2PO_4$, 12 mM $Na_2SO_4$, 0.2% glucose, 0.5% casamino acids, 0.5% agar). The plates were incubated overnight at 37 °C and imaged after 24 h.

## *P. aeruginosa* strain construction

*P. aeruginosa* clinical strains were obtained from the New York State Department of Health Bacteriology Laboratory at the Wadsworth Center. Standard cloning and molecular biology techniques were used to generate plasmids for *E. coli* and *P. aeruginosa* conjugation experiments. Briefly, mutagenesis was performed using a pEXG2 suicide vector containing the coding sequence for the *rhlI* gene plus 500 bp flanking DNA. Conjugative competent *E. coli* SM10 λpir strains harboring pEXG2-*rhlI* mutant plasmids were conjugated with PA14 WT, Δ*lasR* and Δ*lasI* laboratory strains. All strains and plasmids used in this study are shown in Supplementary Table S2. Primers are shown in Supplementary Table S3.

## Whole genome assembly and phylogenetic analyses

Paired-end reads were downloaded from the National Center for Biotechnology Information BioProject database (NCBI accession number: PRJNA288601). Reads were trimmed and assessed for quality using Trim Galore! v0.6.7[50,51]. Trimmed reads were mapped to the *P. aeruginosa* UCBPP-PA14 reference assembly (NCBI accession number: NC_008463.1) using the BWA-MEM[52] algorithm within SAMtools v1.1.0[53]. Variant calling and filtering were performed by BCFtools mpileup v1.1.0.2[54]. Consensus FASTA sequences were generated using BCFtools consensus V1.1.0.2. To identify RhlI SNPs and sequence relatedness, consensus sequences were annotated using Prokka v1.14[55], using default parameters. RhlI-annotated protein sequence alignments were generated using MUSCLE v3.8.1551[56]. Columns with >20% gaps were trimmed and filtered by similarity with trimAl (v1.4.1; option --gt 0.80 and −st 0.001)[57]. The maximum-likelihood tree was generated from RhlI protein sequence alignments ($n = 56$) with IQ-TREE v1.6.12[58]. Model selection was performed using an automatic substitution model based on the Bayesian information criteria (BIC) score, where the HKY +F+I model was chosen. The tree was visualized and annotated with Interactive Tree Of Life (iTOL v. 6.7.1)[59]. To identify RhlI variants among the isolates, we used the *Pseudomonas* Genome DB[60]. We utilized the NCBI BLAST[61,62] search function to compare *rhlI* isolate sequences to the *P. aeruginosa* UCBPP-PA14 reference assembly (NCBI accession number: NC_008463.1). Mismatches were identified using the pairwise output format.

To contextualize RhlI sequence similarity among a diverse range of bacterial species, we obtained protein sequences of RhlI orthologs from the OrthoDB v11[63] online database. Protein sequences were filtered for those which were between 185-215 amino acids. The remaining sequences were aligned using MUSCLE v3.8.1551 and trimmed with trimAl v1.4.1. The maximum-likelihood tree was generated from orthologous RhlI protein sequence alignments ($n = 264$) with IQ-TREE v1.6.12. Model selection was performed using an automatic substitution model based on the Bayesian information criteria (BIC) score, where the LG+F+R7 model was chosen. The tree was visualized and annotated with Interactive Tree Of Life (iTOL) v6.7.1. Custom code and all commands issued in this study can be found at https://github.com/calebmallery/RhlI_manuscript_methods.

## High-resolution mass spectrometry analysis

An ultra-high-performance liquid chromatography (UHPLC) coupled with a high-resolution mass spectrometer (HRMS) method was used for the cell-free supernatant analysis during the time-course study. A high-throughput and direct sample preparation method was developed to make it possible to prepare 100 samples in 2 h, enabling rapid quantitation of the time course study samples. Briefly, 50 μL filtered supernatants were directly extracted by 200 μL methanol and vortexed for 15 s. The precipitated proteins were separated from the extraction solvent using centrifugation (2 min, 8000–10,000×$g$), the supernatant was mixed with 0.1% formic acid in water in an HPLC vial with a 1:1 ratio, and then directly used for UHPLC-HRMS analysis. For UHPLC-HRMS high-throughput analysis, the instrumental system contained a Vanquish UHPLC coupled with a high-resolution QE-Orbitrap mass spectrometer (ThermoFisher Scientific) operating in the positive electrospray-ionization (ESI) mode with a heated ion source (HESI). Analyte separation was achieved using a C18 bonded beads (Hypersil GOLD column, 50 × 2.1 mm, 1.9 μm particle size, ThermoFisher) as the stationary phase and solvent A and B as mobile phases. For gradient elution from the UHPLC, mobile phase A was 0.1% v/v formic acid with 5 mM ammonium formate in water, and mobile phase B was 0.1% v/v formic acid with 5 mM ammonium formate in methanol. The flow rate was 0.450 mL/min, and the column oven was maintained at 45 °C. The binary gradient was initially at 5.0% mobile phase B and increased to 100% mobile phase B over 3.5 min. After a hold at 100% mobile phase B for 1 min, the mobile phase composition was returned to 5% mobile phase B in 0.1 min and maintained for another 1 min to equilibrate the column. The sample injection volume was 2 μl. An external calibration curve was used for quantifying the analytes with quality control samples to ensure the system stability was qualified for each batch. For quantification, a 5 ppm accurate mass window was used for peak integration. Data were acquired using Xcalibur 4.1 software and processed with TraceFinder 5.0 software (both from ThermoFisher).

## RNA extraction and qRT-PCR

RNA extraction and subsequent qRT-PCR experiments on cell pellets from time course assays were performed as described previously[18]. Briefly, pellets were thawed and resuspended in 600 μL TRI Reagent (Sigma) and transferred to a 2 mL screw-cap tube containing 100 μL zirconia beads. Cells were disrupted with a bead beater and 100 μL chloroform was added. Samples were vigorously shaken by hand, incubated at RT for 2 min, and then pelleted at maximum speed for 15 min at 4 °C. The aqueous layer was aspirated and transferred to a clean 1.5 mL microcentrifuge tube. Isopropanol was added to precipitate the RNA. Samples were washed once with 70% ethanol, dried for 2 min, and resuspended in 100 μL nuclease-free water. Resuspended samples were DNase treated using the TURBO DNase kit according to the manufacturer's instructions (ThermoFisher). DNA libraries were prepared with the Superscript IV kit (ThermoFisher) using 500 ng of DNase-treated RNA. SYBR 2x Master Mix (ThermoFisher) was used to perform qRT-PCR on 2 μL of cDNA (diluted 1:5) in a total reaction volume of 20 μL.

## *Galleria mellonella* infection experiments

*P. aeruginosa* strains were cultured from freezer stocks in 3 mL LB overnight at 37 °C with shaking. To prepare larvae inoculant, LB

cultures were back-diluted 1:10 in FDS- media and incubated at 37 °C overnight, then normalized to $10^6$ CFU in PBS. For fast kill assays, late instar larvae (Carolina Biological Supply Company) of uniform size and color were selected for infection. Larvae were injected between the third and fourth proleg with 20 μl of inoculum or PBS control using a BD Micro-Fine IV U-100 Insulin Needle and monitored for 10 h. Death was determined by melanization and failure to respond to physical probing.

## Statistics and reproducibility

All statistical analyses were performed using GraphPad Prism 9.5.1. All experiments were performed in technical duplicate with at least three independent biological replicates performed on different days. Sample sizes were chosen based on previous studies that conducted pyocyanin and C4HSL measurements. No statistical method was used to predetermine the sample size. No data were excluded from the analyses. The experiments were not randomized. The investigators were not blinded to allocation during experiments and outcome assessment.

## Reporting summary

Further information on research design is available in the Nature Portfolio Reporting Summary linked to this article.

## Data availability

The *P. aeruginosa* UCBPP-PA14 reference assembly has a National Center for Biotechnology Information accession number of NC_008463.1. Paired-end reads from clinical strains of *P. aeruginosa* were downloaded from the National Center for Biotechnology Information BioProject database NCBI accession number: PRJNA288601. Individual accession numbers for each sequenced isolate can be found in Supplementary Table S1. Source data are provided with this paper.

## Code availability

Custom code and all commands issued in this study can be found at https://github.com/calebmallery/RhlI_manuscript_methods.

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

## Acknowledgements

This work was supported by National Institutes of Health training grant T32GM132066 to C.P.M. and S.S., NIH grant R01GM14436101, New York Community Trust Foundation grant P19-000454, Cystic Fibrosis Foundation grant PACZKO21G0, and American Lung Association Innovation Award INALA2023 to J.E.P. The authors thank Dr. Kimberlee Musser, Clinical Director of the Wadsworth Center, New York State Department of Health, and the staff in the Bacteriology Group at the Wadsworth Center, for access to clinical isolates of *P. aeruginosa* and their genome sequences. The Wadsworth Center serves as the Northeast AR Lab Network regional laboratory for antimicrobial resistance surveillance. The authors also thank the research laboratories in the Division of Genetics at the Wadsworth Center for helpful discussions on the

research and for resource sharing. This work was made possible with the help of the dedicated staff scientists at the Advanced Genomics Technologies Center and Media & Tissue Core facilities at the Wadsworth Center.

## Author contributions

K.A.S. and J.E.P. conceived the study and designed experiments. K.A.S. performed the experiments shown in Figs. 1e, 2a–d, 3a–d, 4a–d, Supplementary Figs. S3a–d, S4a–e, S5a–d, S6a–d, and S7a–d. M.L.S. performed the experiment shown in Fig. 5. C.P.M. provided bioinformatic analyses displayed in Fig. 1a, Supplementary Table S1, Supplementary Figs. S1a and S2. S.S. performed the experiment shown in Fig. 2e. L.L. provided technical assistance and analyzed all mass spectrometry experiments shown in Fig. 3, Supplementary Figs. S4 and S5. J.E.P. performed structural analyses of the RhlI synthase and homologs. K.A.S., M.L.S., C.P.M., L.L., and J.E.P. wrote the manuscript. J.E.P. supervised the project.

## Competing interests

The authors declare no competing interests.
