## [Peer Review File · Nature Communications]

REVIEWER COMMENTS

Reviewer #1 (Remarks to the Author):

This manuscript by Simanek and colleagues reports on coding variants of rhII in *P. aeruginosa* and how these coding variants might impact QS in clinical isolates. It's interesting conceptually but there are several issues to address.

1. It's not at all clear that the variants they identified arise in the context of infections. There might simply be an alternative wild type in the environment. There is a large collection of sequenced *P. aeruginosa* isolates and strains in public databases and this could be relatively easily determined.
2. The authors' statement that C6-HSL binds with greater affinity to RhIR than C4-HSL was surprising (lines 54-55). Reference 14, which they cite in support of this contention, suggests that C4-HSL has something like 5 or 6x the affinity.
3. Lines 65-73. There's not much evidence in the literature that lasR mutants are actually cheaters in the context of infections and there is some data to the contrary (eg, PMID: 24866798)
4. A citation (or citations) is needed for the statements in lines 79-82. This is also true for the C4-HSL dependency statement in lines 195-196
5. Figures 3 and 4. Why do the authors bury the actual timepoints in the materials rather than label the figures with times?
6. Figure 4. What is the evidence that the variants "optimize" C4-HSL levels? It could be tested experimentally with a rhII mutant and addition of different amounts of C4-HSL. This is important because the authors are making the counterintuitive argument that less signal causes increased transcription of virulence factors, which doesn't have a lot of precedent amongst LuxR homologs
7. Similarly, in figure 4, it's not clear that there actually is a big difference in transcription over the time course, except (again counterintuitively) for rhII -- can the authors explain why more rhII transcription leads to less signal?

8. Figure 5. It is very difficult to make out which line is which The relevant comparison (I think it is Δ lasR versus the Δ lasR - D83E variant) needs a statistical analysis

Reviewer #2 (Remarks to the Author):

P. aeruginosa is a human opportunistic pathogen. It can cause difficult-to-treat chronic infections. *P. aeruginosa* evolves in such infections and it is commonly observed that mutants that are deficient for their main quorum-sensing (QS) system (Las-system) arise and spread. It has long been unclear why a pathogen should lose one of its key systems regulating virulence. Recent findings suggest that the mutations in the Las-system lead to a rewiring of the entire QS-regulon and not a complete loss of function. Specifically, the mutants seem able to decouple the Las-regulon (loss of function) from the subordinate Rhl- and PQS-regulons. The work by Simanek and colleagues builds on this new view of QS evolution. Starting with a phylogenetic analysis, they identified several putative compensatory mutations in the *rhlI* gene (encoding the signal synthetase of the RhlI QS-molecule) in Las-deficient clinical isolates. They then introduced three of these mutations in the a laboratory Las-deficient strain and studied the effect of these mutations on QS-trait activities. They found that these compensatory mutations indeed decouple the Las from the Rhl system, restoring Rhl-activity in a Las-negative background.

Overall, the paper is well written and easy to follow. The experiments are well conducted and the genetic and phenotypic characterisations of the RhlI mutations are novel and insightful. While this paper furthers our understanding of the genetic basis of QS-evolution in *P. aeruginosa*, I am (for now) not convinced that it allows us to reconstruct the biological functions of the investigated compensatory mutations.

Main comments

1. My biggest concern is that all the experimental work was carried out with standard laboratory strains and not clinical isolates. I understand that from a molecular perspective it makes sense to introduce the *rhlI*-mutations in an engineered *lasR*-negative background. It allows to demonstrate the precise effects of these mutations. However, the question of whether the clinical isolates with these *rhlI* mutations show the same phenotypes remains unanswered. This information is absolutely key. Phenotypic data showing that pyocyanin, swarming and virulence are restored in Las-deficient clinical isolates with *rhlI* compensatory mutations is a must to be able to assess the functional consequences of QS-decoupling in infections.

2. It is fashionable to talk about QS-rewiring and decoupling in the context of QS-evolution. However, there is also good evidence that certain Las-mutations lead to a complete functional loss of all QS-systems and traits (e.g. Jiricny et al. 2014 PLoS One; Jayakumar et al. 2022 mSystems). This knowledge tends to get lost. Both loss of function and rewiring can occur, suggesting that there are multiple selection pressures. These aspects must be discussed in a balanced way.

3. I question the usefulness of the Galleria infection experiments in Fig. 5 (see also point 12 below for technical comments). The experiments demonstrate that the compensatory mutations in rhII restore virulence to the wild type level in an acute infection system. This is interesting and okay. However, QS-regulon evolution occurs in the context of chronic infections. It is indisputable that *P. aeruginosa* changes from an invasive acute, to a less virulent biofilm type of lifestyle. QS-regulon evolution is likely involved in this transition. The problem with the Galleria experiments is that it tests for virulence in an highly acute model system. This is far away from the chronic infection situation. The restoration of virulence is probably an artefact of the host system and does not inform us on why QS-regulon modification was selected for. A more likely explanation for the restoration of pyocyanin production is that this molecule is important for redox reactions under low-oxygen conditions in biofilms (see the seminal work by the Lars Dietrich lab). In my opinion, the authors are too focused on the restoration of virulence. Extra experiments (involving biofilm models) and a more balanced discussion on the *P. aeruginosa* lifestyle in chronic infections would be required to better understand the role of rhII compensatory mutations.

Additional comments:

4. Line 47: QS also regulates many self-directed non-collective traits.

5. Line 52: Please specify that the transcription factor is a LasR-LasI dimer.

6. Line 70 and other places: the paper would benefit from a more nuanced view on 'social cheaters'. Cheating is one explanation for the selection of QS-negative mutants. The rewiring hypothesis actually speaks against it. QS-mutants are selected for because QS-rewiring is beneficial itself and not because of social exploitation.

7. Line 119: This statement is strange. Mutations are always stochastic. Enrichment of specific mutations is a sign of natural selection and this process is non-stochastic. Please revise accordingly.

8. Line 137: Please list the relevant studies. How many studies reported such mutants and how frequent were these mutants?

9. Figure 2: Please add labels to the individual panels so that the reader can immediately link the panels to the respective strains and mutants. Moreover, the linear line fits for the mutants make little sense. The data clearly show that the relationship between OD and pyocyanin production is exponential. Please use corresponding curve fits.

10. Line 202: The statement “The expression of rhlA ... was abolished in a WT strain” is misleading. The expression declined but was not abolished.

11. Figure 4: Please add the actual time in hours on the x-axis. Time point 1-4 means little. Furthermore, add gene labels to each panel.

12. Galleria experiments: Why did the authors use 10^6 cells to infect Galleria. *P. aeruginosa* kills larvae even with 10^1 cells. The advantage of lower infectious doses is that there can be more distinct differences in the survival curves between mutants. Related to this, statistical analyses are required to show that there are indeed significant differences between mutants.

13. Line 237: This paper features no data showing that rhlI compensatory mutations confer a fitness advantage in vivo. Please revise.

14. Figure 5B: There are different shades of green and it is unclear what they mean. Please add a color code to the figure itself.

Reviewer #3 (Remarks to the Author):

The authors describe in their communication how *P. aeruginosa* can cope with mutations of the autoinducer master-regulator LasR by using Rhl for maintaining virulence. This is an important aspect that might help to understand the ecology of virulence of *P. aeruginosa*. Furthermore, from a functional perspective the suppressive activity of C4-HSL to pyocyanin production is noteworthy.

The manuscript is well written and the experiments are well described and well-planned. Various experiments supporting the major results from different aspects are used that combine to a comprehensive result. The Discussion could be shortened a bit and be a bit more focussed. I am

A figure in the introduction about the complex interaction of the Las and Rhl-systems would help the reader who is not a specialist in autoinducer signaling. Furthermore, pyocyanin is not introduced in the introduction. Is it the virulence factor, or only an indicator for it?

In summary, I have not much to complain about this interesting manuscript worth to be published in Nature communications.

L170: what shared precursor could that be? The fatty acid part is largely different in chain length. Sam is ubiquitous in cells.

L179: Systematic name for pyocyanin should be used at first mentioning in the main text.

Reviewer #1 (Remarks to the Author):

This manuscript by Simanek and colleagues reports on coding variants of *rhII* in *P. aeruginosa* and how these coding variants might impact QS in clinical isolates. It's interesting conceptually but there are several issues to address.

1. It's not at all clear that the variants they identified arise in the context of infections. There might simply be an alternative wild type in the environment. There is a large collection of sequenced *P. aeruginosa* isolates and strains in public databases and this could be relatively easily determined.

The mutations identified and characterized are from clinical isolates of *P. aeruginosa*, which are pathogenic strains that have been isolated from patients with infections. To further address the reviewer's comment, we created a dendrogram showing the prevalence of *rhII* mutations in *P. aeruginosa* strains with published sequencing data. The dendrogram is displayed in a new figure, Figure S2. The mutations correlate to the isolates, which are noted as infection isolates in the metadata.

2. The authors' statement that C6-HSL binds with greater affinity to RhIR than C4-HSL was surprising (lines 54-55). Reference 14, which they cite in support of this contention, suggests that C4-HSL has something like 5 or 6x the affinity.

We acknowledge the discrepancy in the literature. C₆HSL more robustly activated signaling in an *E. coli* reporter system, but this was not the case in *P. aeruginosa*. Since we do not address C₆HSL binding in this paper, we have removed this sentence from the manuscript.

3. Lines 65-73. There's not much evidence in the literature that *lasR* mutants are actually cheaters in the context of infections and there is some data to the contrary (eg, PMID: 24866798)

This information has been added to the manuscript at lines 102-105.

4. A citation (or citations) is needed for the statements in lines 79-82. This is also true for the C4-HSL dependency statement in lines 195-196

We have added citations to this statement.

5. Figures 3 and 4. Why do the authors bury the actual timepoints in the materials rather than label the figures with times?

We have changed the axes labels to reflect the actual time at which the samples were collected.

6. Figure 4. What is the evidence that the variants "optimize" C4-HSL levels? It could be tested experimentally with a *rhII* mutant and addition of different amounts of C4-HSL.

This is important because the authors are making the counterintuitive argument that less signal causes increased transcription of virulence factors, which doesn't have a lot of precedent amongst LuxR homologs

We performed this experiment as suggested with a $\Delta lasR$ D83E strains and added the data to Figure 3E. We show that addition of exogenous C₄HSL repressed pyocyanin production in this strain. We also performed a C₄HSL dose-response on clinical strains and show that exogenous C₄HSL represses pyocyanin compared to the respective untreated culture for each strain (Figure S7).

7. Similarly, in figure 4, it's not clear that there actually is a big difference in transcription over the time course, except (again counterintuitively) for *rhII* -- can the authors explain why more *rhII* transcription leads to less signal?

We think the referee might be mistaken. The expression of *rhII* is largely the same across all strains and statistical comparisons were deemed insignificant for *rhII* expression. We maintain that *rhIA* transcription changes due to the levels of C₄HSL produced by each strain, not the transcription of *rhII* itself.

8. Figure 5. It is very difficult to make out which line is which The relevant comparison (I think it is $\Delta lasR$ versus the $\Delta lasR$ - D83E variant) needs a statistical analysis

We have simplified the graph in Figure 5 to separate out the controls and the mutants. With all the data now displayed in Figure 5, we have removed the former Figure S6 from the manuscript. We have added statistical analyses of mutant strains compared to the $\Delta lasR$ per the referee's suggestions.

Reviewer #2 (Remarks to the Author):

P. aeruginosa is a human opportunistic pathogen. It can cause difficult-to-treat chronic infections. *P. aeruginosa* evolves in such infections and it is commonly observed that mutants that are deficient for their main quorum-sensing (QS) system (Las-system) arise and spread. It has long been unclear why a pathogen should lose one of its key systems regulating virulence. Recent findings suggest that the mutations in the Las-system lead to a rewiring of the entire QS-regulon and not a complete loss of function. Specifically, the mutants seem able to decouple the Las-regulon (loss of function) from the subordinate Rhl- and PQS-regulons. The work by Simanek and colleagues builds on this new view of QS evolution. Starting with a phylogenetic analysis, they identified several putative compensatory mutations in the *rhII* gene (encoding the signal synthetase of the Rhl QS-molecule) in Las-deficient clinical isolates. They then introduced three of these mutations in a laboratory Las-deficient strain and studied the effect of these mutations on QS-trait activities. They found that these compensatory mutations indeed decouple the Las from the Rhl system, restoring Rhl-activity in a Las-negative background.

Overall, the paper is well written and easy to follow. The experiments are well

conducted and the genetic and phenotypic characterisations of the RhlI mutations are novel and insightful. While this paper furthers our understanding of the genetic basis of QS-evolution in *P. aeruginosa*, I am (for now) not convinced that it allows us to reconstruct the biological functions of the investigated compensatory mutations.

Main comments

1. My biggest concern is that all the experimental work was carried out with standard laboratory strains and not clinical isolates. I understand that from a molecular perspective it makes sense to introduce the rhl-mutations in an engineered *lasR*-negative background. It allows to demonstrate the precise effects of these mutations. However, the question of whether the clinical isolates with these rhlI mutations show the same phenotypes remains unanswered. This information is absolutely key. Phenotypic data showing that pyocyanin, swarming and virulence are restored in *Las*-deficient clinical isolates with rhlI compensatory mutations is a must to be able to assess the functional consequences of QS-decoupling in infections.

The reviewer's comment is well-taken. We have included additional data on acute infection isolates and show that strains which have concurrent *lasR* truncations and *rhlI* mutations synthesize less C₄HSL, produce more pyocyanin, and upregulate *rhlA* compared to a clinical strain with WT *lasR* and *rhlI*.

2. It is fashionable to talk about QS-rewiring and decoupling in the context of QS-evolution. However, there is also good evidence that certain *Las*-mutations lead to a complete functional loss of all QS-systems and traits (e.g. Jiricny et al. 2014 PLoS One; Jayakumar et al. 2022 mSystems). This knowledge tends to get lost. Both loss of function and rewiring can occur, suggesting that there are multiple selection pressures. These aspects must be discussed in a balanced way.

We disagree with the reviewer's comment. Both papers that the reviewer cited in their response demonstrate a loss of *LasR* associated virulence over time but discuss how RhlR QS is maintained or rewired. Therefore, the loss of *LasR* function and rewiring of RhlR occur simultaneously, not separately. Our own data shows that RhlR mediated QS traits are maintained, albeit delayed, in a *LasR*- strain and are consistent with prior observations.

3. I question the usefulness of the *Galleria* infection experiments in Fig. 5 (see also point 12 below for technical comments). The experiments demonstrate that the compensatory mutations in rhlI restore virulence to the wild type level in an acute infection system. This is interesting and okay. However, QS-regulon evolution occurs in the context of chronic infections. It is indisputable that *P. aeruginosa* changes from an invasive acute, to a less virulent biofilm type of lifestyle. QS-regulon evolution is likely involved in this transition. The problem with the *Galleria* experiments is that it tests for virulence in an highly acute model system. This is far away from the chronic infection situation. The restoration of virulence is probably an artefact of the host system and does not inform us on why QS-regulon modification was selected for. A more likely explanation for the restoration of

pyocyanin production is that this molecule is important for redox reactions under low-oxygen conditions in biofilms (see the seminal work by the Lars Dietrich lab). In my opinion, the authors are too focused on the restoration of virulence. Extra experiments (involving biofilm models) and a more balanced discussion on the *P. aeruginosa* lifestyle in chronic infections would be required to better understand the role of *rhII* compensatory mutations.

The reviewer is correct that *lasR* mutations are selected for during chronic infection, and these strains are less virulent. Here, we are characterizing *rhII* mutations in acute infection isolates. While the *rhII* mutations don't seem to change the phenotypes of a strain with wild type *lasR*, virulence is enhanced in *lasR* deletion strains. Just like *P. aeruginosa* transitions from an acute to chronic lifestyle, it must switch back to acute/invasive to remain infective in a hospital niche (establishing infection in a patient that is exposed to colonized medical equipment or another infected patient). It's possible that *rhII* mutations restore virulence in a *lasR* deficient strain to restore an acute phenotype and perpetuate infection of new hosts. Here, we use a *Galleria mellonella* infection model to demonstrate acute infection, not chronic infection.

Additional comments:

4. Line 47: QS also regulates many self-directed non-collective traits.

We agree that there is some nuance to QS that is glossed over with this statement and have included "individualistic behaviors" in the main text on line 55.

5. Line 52: Please specify that the transcription factor is a LasR-LasI dimer.

We believe the referee might be mistaken. LasI synthesizes the autoinducer 3OC₁₂HSL, which binds the transcription factor LasR. LasI does not bind LasR. However, we now clarify that LasR dimerizes upon activation by 3OC₁₂HSL.

6. Line 70 and other places: the paper would benefit from a more nuanced view on 'social cheaters'. Cheating is one explanation for the selection of QS-negative mutants. The rewiring hypothesis actually speaks against it. QS-mutants are selected for because QS-rewiring is beneficial itself and not because of social exploitation.

We have added text and cited additional research with alternative explanations for the selection of QS negative mutants in infections.

7. Line 119: This statement is strange. Mutations are always stochastic. Enrichment of specific mutations is a sign of natural selection, and this process is non-stochastic. Please revise accordingly.

We agree that the statement was unclear. We have revised the wording to state that the mutations characterized in this study are consistent in clinical isolates, suggesting that there is a functional advantage to maintaining these mutations.

8. Line 137: Please list the relevant studies. How many studies reported such mutants and how frequent were these mutants?

We apologize for the lack of clarity; we are not referencing other studies here. We are referring to the cohort of isolates we analyzed to identify mutations in *rhII*. The frequency of the mutations in *P. aeruginosa* strains is described in Figure 1A, Figure S2, and Table S1, and we have cited this in the text for clarity.

9. Figure 2: Please add labels to the individual panels so that the reader can immediately link the panels to the respective strains and mutants. Moreover, the linear line fits for the mutants make little sense. The data clearly show that the relationship between OD and pyocyanin production is exponential. Please use corresponding curve fits.

We have added labels to the panels to improve the reader's ability to link the data with the appropriate strain. We appreciate the suggestion to change from linear regression analyses to non-linear regression analyses for the correlation data in Figure 2 and Figure S3 because it significantly improves the visualization of the data and our interpretation of the role of RhII variants in both WT and $\Delta lasR$. We have changed all graphs accordingly in Figure 2 and Figure S3.

10. Line 202: The statement "The expression of rhIA ... was abolished in a WT strain" is misleading. The expression declined but was not abolished.

We changed the wording to "declined" instead of "abolished".

11. Figure 4: Please add the actual time in hours on the x-axis. Time point 1-4 means little. Furthermore, add gene labels to each panel.

We changed the x-axis labels per the reviewer's suggestion.

12. Galleria experiments: Why did the authors use 10^6 cells to infect Galleria. *P. aeruginosa* kills larvae even with 10^1 cells. The advantage of lower infectious doses is that there can be more distinct differences in the survival curves between mutants. Related to this, statistical analyses are required to show that there are indeed significant differences between mutants.

We chose to inoculate with 10^6 cells to ensure infection efficiency. The reviewer's suggestion to use a lower infectious dose is well-taken, but we chose not to repeat the experiment with 10^1 cells for this paper because our initial experiment already shows that *rhII* mutants kill larvae at a faster rate. We have added statistical analyses as requested.

13. Line 237: This paper features no data showing that rhII compensatory mutations confer a fitness advantage in vivo. Please revise.

We agree. We have removed this statement from the manuscript.

14. Figure 5B: There are different shades of green and it is unclear what they mean. Please add a color code to the figure itself.

Thank you for this helpful suggestion. We have added a color-coded legend as suggested.

Reviewer #3 (Remarks to the Author):

The authors describe in their communication how *P. aeruginosa* can cope with mutations of the autoinducer master-regulator LasR by using Rhl for maintaining virulence. This is an important aspect that might help to understand the ecology of virulence of *P. aeruginosa*. Furthermore, from a functional perspective the suppressive activity of C4-HSL to pyocyanin production is noteworthy.

The manuscript is well written and the experiments are well described and well-planned. Various experiments supporting the major results from different aspects are used that combine to a comprehensive result. The Discussion could be shortened a bit and be a bit more focused.

A figure in the introduction about the complex interaction of the Las and Rhl-systems would help the reader who is not a specialist in autoinducer signaling. Furthermore, pyocyanin is not introduced in the introduction. Is it the virulence factor, or only an indicator for it?

We plan to add a schematic of quorum sensing signaling in a graphical abstract. We have added a description of pyocyanin in the introduction at line 66.

In summary, I have not much to complain about this interesting manuscript worth to be published in Nature communications.

L170: what shared precursor could that be? The fatty acid part is largely different in chain length. Sam is ubiquitous in cells.

We now specify what the shared precursors are for RhII and LasI on line 386-387.

L179: Systematic name for pyocyanin should be used at first mentioning in the main text.

Now that we mention the role of pyocyanin in the introduction, related to comment #1, we provide the systematic name for pyocyanin there.

REVIEWERS' COMMENTS

Reviewer #1 (Remarks to the Author):

The authors have addressed the concerns I expressed in my initial review. The current version is substantially improved and considerably better in terms of presentation and logic.

Reviewer #2 (Remarks to the Author):

The authors have carefully revised their manuscript. Particularly, the extra experiments conducted with clinical isolates are important. I understand that data from clinical isolates are typically noisy, given that such isolates differ in many aspects other than their QS mutations. Nonetheless, the inclusion of such data increases the relevance of the paper.

I have one small extra comment. Certain labels in some of the figures are very small (e.g. Fig. 1A, 1E, 3A-D, 5A) and hard to read. I recommend to enlarge labels when preparing the final figure versions.

REVIEWERS' COMMENTS

Reviewer #1 (Remarks to the Author):

The authors have addressed the concerns I expressed in my initial review. The current version is substantially improved and considerably better in terms of presentation and logic.

We thank the referee for their assistance in enhancing the quality of the manuscript.

Reviewer #2 (Remarks to the Author):

The authors have carefully revised their manuscript. Particularly, the extra experiments conducted with clinical isolates are important. I understand that data from clinical isolates are typically noisy, given that such isolates differ in many aspects other than their QS mutations. Nonetheless, the inclusion of such data increases the relevance of the paper.

I have one small extra comment. Certain labels in some of the figures are very small (e.g. Fig. 1A, 1E, 3A-D, 5A) and hard to read. I recommend to enlarge labels when preparing the final figure versions.

We thank the referee for their assistance in enhancing the quality of the manuscript. As per their suggestion, we increased the font size for all figure axes labels throughout the manuscript to enhance readability.